# Time Prediction Model for Pointing at Target Having Different Motor and Visual Width with Distractors

**Leave Authors Anonymous**
for Submission
City, Country
e-mail address

**Leave Authors Anonymous**
for Submission
City, Country
e-mail address

**Leave Authors Anonymous**
for Submission
City, Country
e-mail address

## ABSTRACT

In this study, we extend Fitts' law to enable it to predict the movement time of pointing operations in interfaces, such as in navigation bars whose items have different motor and visual widths and intervals between a target and distractors. For this, we conduct two experiments to investigate the presence or absence of the distractors that affect pointing operations and how increasing the size of the intervals changes user performance. We found that the movement time is strongly affected by the motor width and intervals and slightly by the visual width. On the basis of the results, we constructed a model for considering the difference between the motor and visual widths and the intervals between the target and distractors. The model allows user-interface designers to configure these factors on the basis of movement time. Our model also showed a good fit for not only the data of our two experiments but also those of three previous studies. We also discuss future work for making our model more practical.

## Author Keywords

Difference between motor and visual widths; distractor; pointing; Fitts' law; GUIs.

## CCS Concepts

•**Human-centered computing** → *HCI theory, concepts and models;*

## INTRODUCTION

### Background

In graphical user interfaces (GUIs), users move a cursor and then click on a desired object (*target*), e.g., for opening a file, executing an application, or going to another webpage. This is called *pointing* and is one of the fundamental operations in GUIs. The movement time of pointing operations is modeled by using Fitts' law [13, 22]. Fitts' law can be applied to many input devices (e.g., mice [13, 30], styli [30], and fingers [9]) and used for predicting the movement times of other operations (e.g., passing through two goals called *crossing* [1]).

*GI '20,* May 21–22, 2020, Toronto, Canada

© 2020 Copyright held by the owner/author(s). Publication rights licensed to ACM.
ISBN 978-1-4503-6708-0/20/04. . . $15.00

DOI: https://doi.org/10.1145/3313831.XXXXXXX

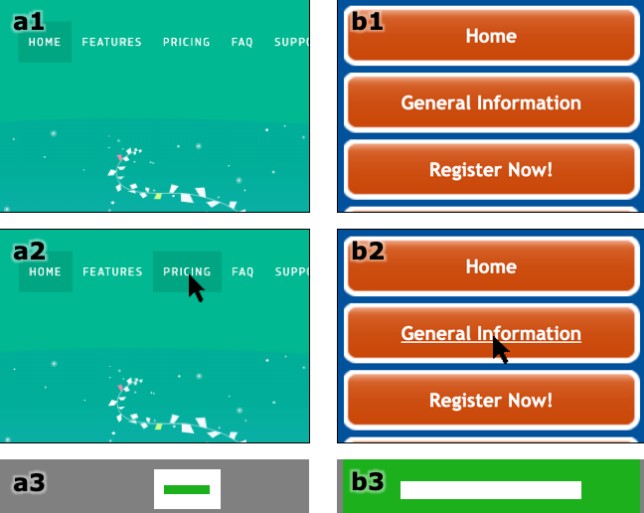

**Figure 1. Two navigation-bar examples. Motor width (white) is (a) larger or (b) smaller than visual width (green). In both navigation bars, a target is surrounded by distractors. In (b), there are intervals between the target and distractors.**

In addition, by combining Fitts' law with other models, the movement time of navigation in a hierarchy menu [4, 18, 31] and selecting multiple objects by using a lasso tool [31, 43] can be predicted. Moreover, Fitts' law has been modified to predict the movement time more accurately for other situations [2, 10, 21, 25, 28, 29, 33]. The above studies (i.e., modeling pointing operations and refining pointing models) have contributed to, for example, evaluating input devices, designing interfaces, and better understanding human motor control.

Before describing our research questions, we give examples of pointing operation: clicking on a target in two navigation bars (Figure 1). In Figure 1a[1], users want to go to another page, so they click an item ("PRICING") in the navigation bar (a1). At this time, the users may aim at the item's text because it is unclear where the item is clickable (they may believe that the text at least is clickable). However, when their cursors enter the clickable area, the area is highlighted in dark color (a2), so the users realize that the clickable area is larger than the item's text (a3). That is, the users can click not only the item's text but also its surrounding area. In this paper, we define the target's clickable area as the *motor width* and the area displayed on the screen as the *visual width*. In this

---

[1] https://www.stillio.com/

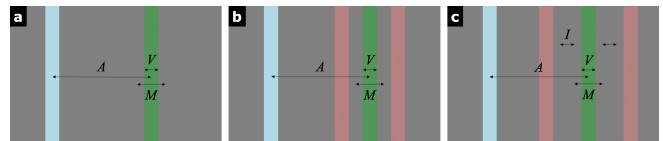

**Figure 2. (a) Previous study's task [35, 36]. (b) Our Experiment 1 task. (c) Our Experiment 2 task. A is distance to target, M is motor width, V is visual width, and I is interval between target and distractors. In (a), participants must click blue start area and then green end area. In (b) and (c), they must click blue start area and then motor width of green end area.**

GUI, the motor width is larger than the visual width (item's text). In contrast, Figure 1b[2] shows GUI in which the motor width is smaller than the visual width. Users aim at the whole orange button but can click only the item's text ("General Information").

In summary, in GUIs, the motor and visual widths are often different. Of course, GUIs also exist in which the motor width equals the visual width. In addition, the target as shown in Figure 1 is sandwiched by *distractors* that users do not want to click. The distractors have similar appearance, motor width, and visual widths to the target. Thus, the users have to point at the target successfully while avoiding the distractors. Moreover, the intervals between the target and distractors do not exist (Figure 1a) or do exist (Figure 1b).

**Research Questions and Key Contribution**
Fitts' law can predict the movement time ($MT$) on the basis of the target width ($W$) and distance ($A$) to the target (Equation 3). As shown in Figure 1, the target has two widths: the motor and visual widths. However, it is not defined whether $W$ in Fitts' law indicates the motor or visual width. Thus, the range where Fitts' law is used is limited to a situation where the motor and visual widths are equal. In this study, we extend Fitts' law to enable to it predict the movement time even when the motor and visual widths are different. In addition, our model can also consider the intervals between the target and distractors. That is, the designers can also adjust navigation bars such those in as Figure 1 on the basis of a quantitative model.

In this study, we conduct two experiments to investigate (1) how the presence or absence of the distractors affects pointing operations (Figure 2b) and (2) how the users' performance changes when the intervals between the target and distractors are enlarged (Figure 2c). On the basis of the experimental results, we build a model for considering the difference between motor and visual widths and the intervals. As a short summary, we introduce a model built in this paper (for detailed modeling, see Experiments 1 and 2). The model (Equation 1) considers the difference between the motor and visual widths.

$$MT = a + b \log_2 \left( \sqrt{\left(\frac{A}{M}\right)^2 + c \left(\frac{A}{V}\right)^2} + 1 \right) \quad (1)$$

where $MT$ is movement time, $A$ is the distance to a target, $M$ is the motor target width, and $V$ is the visual target width with

[2] https://web.archive.org/web/20110308051632/http://www.asaging.org/aia11/

three regression constants ($a$, $b$, and $c$). This model well fits not only the data of our experiment but also the data of three previous studies. When the motor width equals the visual width (i.e., this is a normal Fitts' task, $M = V$), this model is mostly consistent with Fitts' law.

By adding an additional regression constant ($d$) and the term including the intervals ($I$) between the target and distractors, the model (Equation 2) can consider the intervals. The final form of our model is consistent with Equation 1, e.g., when $I = 0$ (i.e., intervals do not exist), the term including $I$ can be merged with $a$.

$$MT = a + b \log_2 \left( \sqrt{\left(\frac{A}{M}\right)^2 + c \left(\frac{A}{V}\right)^2} + 1 \right) + d \log_2 \left(\frac{1}{I+0.0049} + 1\right) \quad (2)$$

In our two experiments, we found that using the effective width [20, 29, 33] also showed sufficient fits. That is, researchers can fairly compare input devices that have different pointing accuracies even in situations such as those in Figure 1.

## RELATED WORK

### Pointing Model
Fitts' law [13, 22] is a pointing model for predicting the movement time ($MT$) of pointing at the target that has distance ($A$) from a point and width ($W$). This model can be expressed as follows:

$$MT = a + b \log_2 \left(\frac{2A}{W}\right)$$
$$= a' + b \log_2 \left(\frac{A}{W}\right) \quad (3)$$

where $a$ and $b$ are regression constants, and $a' = a + b \log_2 2$. We use the lower row in Equation 3 as (an equivalent version of) the original Fitts' law. The logarithm term in Fitts' law is called *index of difficulty* ($ID$), i.e., increasing $ID$ increases the predicted $MT$. A high $ID$ means an interface in which users have difficulty performing pointing operations, i.e., long $MT$ is needed. There are many versions of Fitts' law [21, 25], and the Shannon formulation [28] (adding '+1' to the original Fitts' law, Equation 4) has been known to show a better fit.

$$MT = a + b \log_2 \left(\frac{A}{W} + 1\right) \quad (4)$$

If different two input devices are compared, one would find that one device is faster but less accurate and the other is slower but more accurate. Thus, it is difficult to answer the question of which device performs better. In such a case, researchers use the effective width that can adjust the error rates of the input devices to make them the same [20, 29, 33], which allows them to compare the two devices assuming that the devices have the same accuracy. The effective width ($W_e = \sqrt{2\pi e}\sigma$) is calculated by using the standard deviation ($\sigma$) of clicked endpoints; $W$ in $ID$ is replaced with $W_e$, and the index of difficulty is called $ID_e$ (Equation 5).

$$ID_e = \log_2 \left(\frac{A}{W_e} + 1\right), \quad (5)$$

Using the effective amplitude $A_e$ instead of A in Equation 5 can adjust the distance. However, the effect of $A_e$ is smaller than that of $W_e$ [45]. Thus, we use $ID_e$ based only on $W_e$.

In normal Fitts' tasks, the target has a certain width and practically infinite height, i.e., a 1D pointing task. However, in actual GUIs, targets have finite width and height, i.e., the target is often rectangular, and this is a 2D pointing task. There are many 2D pointing models, and we give one example (Equation 6) [19].

$$MT = a + b\log_2\left(\frac{A}{W}\right) + c\log_2\left(\frac{A}{H}\right) \qquad (6)$$

where $H$ is the height of the target and $c$ is an additional regression constant. This model means that $W$ and $H$ independently affect $MT$. However, Accot and Zhai [2] later found the interaction for $W \times H$ on $MT$. Thus, the model is modified as follows:

$$MT = a + b\log_2\sqrt{\left(\frac{A}{W}\right) + \eta\left(\frac{A}{H}\right)} \qquad (7)$$

where $\eta$ is the free weight. In Equation 6, when $c$ is smaller than $b$, Equation 6 can be approximated as Equation 7 [26].

Blanch et al. [10] defined the index of sparseness ($IS$, Equation 8) by using space with the distractors ($\rho$).

$$IS = \log_2\frac{1}{\rho} \qquad (8)$$

$\rho$ is between 0 and 1. When $\rho = 1$, there is no space between the target and distractors, and when $\rho$ is decreased, the space between the target and distractors is increased. In addition, the movement time considering the space can be expressed as follows:

$$MT = a + bID - cIS \qquad (9)$$

That is, increasing the space between the target and distractors (decreasing $\rho$) means decreasing the movement time.

### Difference between Motor and Visual Sizes of Target

Usuba et al. investigated the effect of the difference between the motor and visual widths on mouse pointing operations through two studies: (1) a situation in which the target is small, such as window frames [36], and (2) a situation in which the target is larger than in the previous study such as items in navigation bars [35]. In both studies, the movement time strongly depended on motor width, and although the effect of visual width is not significant, increasing it decreases movement time. In addition, $\sigma$ depends on motor width; thus, the effective width shows a good fit [35]. However, as noted in previous studies [15, 39, 45], because only the nominal width is informative for UI designers, the effective width should be used, e.g., when comparing the performance of input devices when participants' pointing precision varies. Thus, a model is needed that can predict the movement time in a situation where the difference between the motor and visual widths exist without using the effective width. In their studies, Usuba et al. did not develop such a model. In addition, they did not consider the effect of the distractors (Figure 2a). Also in touch pointing operations, they examined the effect of the difference between the motor and visual widths [37].

Chapuis and Dragicevic [15] investigated the performance of small target acquisition under several visual and motor scales, e.g., conducting an experiment by only magnifying the appearance of the target by not changing the control-display (C-D) gain, only decreasing it, or using a combination. The C-D gain means the mapping between the physical mouse movement and cursor movement on the display, and when the C-D gain is decreased, users only slightly move a cursor even if they significantly move the mouse. Thus, changing the C-D gain allows users to feel as if the motor width is enlarged without changing the visual width. Chapuis and Dragicevic found that increasing the motor scale (decreasing the C-D gain) increased movement time; however, increasing the visual scale (magnifying the target appearance) does not affect movement time much.

Area-cursor techniques [16, 23, 27, 34, 38] expand an activation area where a click event fires on a cursor. Expanding the activation area equals expanding the target size, i.e., this approach equals expanding the motor width. In the Dock of macOS, the icons become larger as a cursor moves closer, which called *target expansion* [17, 24, 44]. In both area-cursor and target expansion techniques, the movement time depends on final target size, i.e., the motor width. On the other hand, in many area-cursor and target expansion techniques, the activation area and target size dynamically change. Thus, the situation focused on in this study, where the motor and visual widths are statically different, has not been explored much.

### Effect of Distractors on Pointing Operations

Blanch et al. [10] investigated and modeled mouse pointing operations with distractors (Equation 9). In touch pointing operations, the effect of the spaces between the target and distractors was also investigated [40–42]; small spaces negatively affect the error rate but do not strongly change the task completion time. A similar tendency in the effect was confirmed in crowd-based experiments [42]. Especially for the touch operations, *placeholder effects* [3, 11] have been known. This effect means that a farther target can be acquired more quickly when items are lined up horizontally. In summary, users' performance of pointing operations depends on the size of the space between the target and distractors and whether the distractors exist.

### EXPERIMENT 1: DISTRACTOR EFFECTS

### Apparatus

We used an Apple MacBook Pro laptop (Intel Core i5, 2.4 GHz, two cores, Intel Iris 1536 MB, 8 GB of RAM, macOS Sierra, Figure 3). The display scaling resolution was 1680 × 1050 pixels (the actual size was 13.3 inches, 286.47 × 179.04 mm, 0.17 mm/pixel resolution). We used an optical gaming mouse, Logitech G-PPD-002WL (3200 dpi), as an input device. The mouse was connected to the laptop with a 1.80-m cable. A large enough mouse pad (899 × 420 mm) was used. The full-screen experimental system was developed with JavaScript.

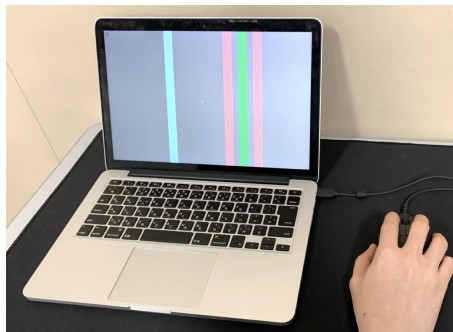

**Figure 3. Experimental equipment.**

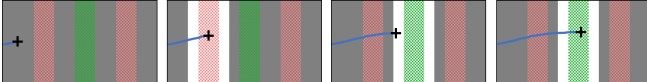

**Figure 4. When cross-hair cursor enters motor width of target or distractor, motor width was highlighted in white. Note that blue trajectory is for explanation; not drawn in actual systems.**

### Participants

Twelve paid volunteers participated in this study (five females and seven males; age: $M = 21.83$, $SD = 1.14$). All participants were right-handed and operated the mouse accordingly. Each participant received the equivalent of US$46 for their time.

### Task

The visual stimuli included a blue start area, red distractors, and a green target (Figure 2b). The participants clicked on the start area to start a trial and aimed for the target as quickly and accurately as possible. At the start of the trial, a start sound was played and measurement began. When a cross-hair cursor entered the motor width of the target or distractor, as with the example navigation bar (Figure 1a), the motor width was highlighted in white (Figure 4)[3]. When the participants clicked within the motor width of the target, a success sound was played. Otherwise, a failure sound was played, and the trial was flagged as an error. Following previous studies [14,15], we asked the participants to avoid *clutching* (replacing the mouse on the mouse pad)[4]. If the participants clutched, they pushed the right button of the mouse and redid the trial. Retrials due to clutching were not regarded as errors.

### Design and Procedure

The *A* from the center of the starting area to the center of the target was 600 or 800 pixels (102.31 or 136.41 mm, respectively). The *M* and *V* were 20, 40, 70, or 120 pixels (3.41, 6.82, 11.94, or 20.46 mm, respectively); motor width was larger than, equal to, or smaller than the visual width. To compare the effects of the distractors, we tested two conditions on the existence of the distractors. When *Distractor = True*, there were the

---

[3]Even if the visual width equaled the motor width, because the visual width was lit by highlighting the motor width, the participants could perceive the highlight.
[4]We did this because clutching may reduce the fitness of pointing models [14]. If we had allowed clutching and obtained poor regression fitness, it would have been unclear whether the results were due to clutching or experimental conditions such as the difference between the motor and visual widths.



**Figure 5. Arrangements of motor and visual widths of target and distractors for three possible conditions.**

red distractors in the task; however, when *Distractor = False*, there were no distractors. We used the same values for *A*, *M*, and *V* as in previous studies [35, 36].

The values of motor and visual widths of the start area equaled *V* because we wanted to prevent the participants from presuming the motor width of the target before starting a trial. The motor and visual widths of the target equaled those of the distractors. There was no margin between the larger of the motor and visual widths (Figure 5).

One *set* consisted of $2A \times 4M \times 4V = 32$ trials for a fixed *Distractor* condition. The orders of *A*, *M*, and *V* were randomized in a set. By each *Distractor*, after an introductory practice set, each participant completed ten sets to produce experimental data. The order of *Distractor* was balanced among the 12 participants. A total of 7,680 trials (i.e., $2Distractor \times 2A \times 4M \times 4V \times 10$ sets $\times 12$ participants) were conducted, which required approximately 20 min per participant.

### Measurements

The dependent variables included the dwell time *DT* (the time from entering the target to clicking the target, excluding error trials), *MT* (the time from clicking the start area to clicking the target, excluding error trials), standard deviation of x-coordinate $SD_x$ (the origin was the center of the target, including the error trials), and error rate. The data processing followed that in previous studies [29, 33, 35, 36].

### RESULTS

Among the 7,678 trials (2 were outliers[5]), 143 errors occurred (1.86%). The error rate was lower than those in previous studies [23, 29, 33, 35]. According to the participants' comments after the experiment, they performed the pointing operation while watching the highlight of the motor width. Thus, we believe that because the highlight allows the participants to operate more accurately, a lower error rate was observed. On the other hand, the fact that the highlight helped in pointing operations was the opposite of the effect found in previous studies [6, 7].

We analyzed the data by using repeated-measures analysis of variations (ANOVA) with Bonferroni correction as the *p*-value adjustment method. The independent variables were *Distractor*, *A*, *M*, and *V*, and the dependent variables were the same as those used in the measurements. In our graphs

---

[5]When the clicked position was below $A/2$, the trial was regarded as an outlier following previous studies [8, 33, 35]. We did not use the criterion based on *W* because this task had different motor and visual widths.

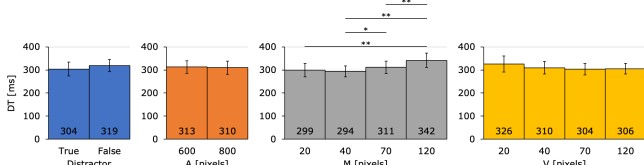

**Figure 6.** *DT* **vs.** *Distractor*, *A*, *M*, **and** *V*.

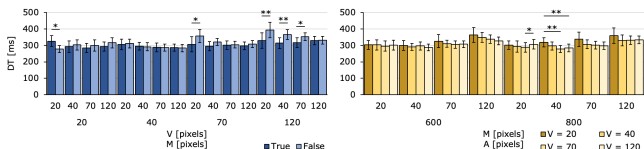

**Figure 7.** *DT* **for** *Distractor* $\times M \times V$ **and** $A \times M \times V$.

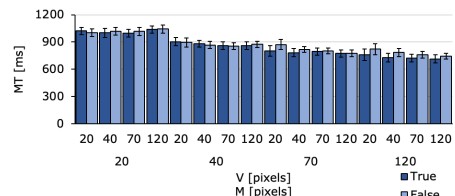

**Figure 9.** *MT* **vs.** *Distractor* $\times M \times V$.

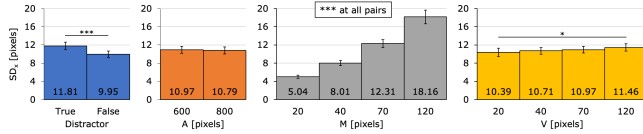

**Figure 10.** *SD$_x$* **vs.** *Distractor*, *A*, *M*, **and** *V*.

of the results, the error bars represent the standard error, and ***, **, and * indicate $p < 0.001$, $p < 0.01$, and $p < 0.05$, respectively.

### Dwell Time
We observed the main effect for *M* ($F_{3,33} = 15.02, p < 0.001, \eta_p^2 = 0.58$) not *Distractor* ($F_{1,11} = 1.78, p = 0.21, \eta_p^2 = 0.14$), *A* ($F_{1,11} = 2.59, p = 0.14, \eta_p^2 = 0.19$), and *V* ($F_{3,33} = 2.59, p = 0.07, \eta_p^2 = 0.19$). Figure 6 shows the results of the post-hoc test. We also observed the interactions for *Distractor* $\times M$ ($F_{3,33} = 11.12, p < 0.001, \eta_p^2 = 0.50$), $A \times V$ ($F_{3,33} = 4.00, p < 0.05, \eta_p^2 = 0.27$), *Distractor* $\times M \times V$ ($F_{9,99} = 11.40, p < 0.001, \eta_p^2 = 0.51$, Figure 7), and $A \times M \times V$ ($F_{9,99} = 2.18, p < 0.05, \eta_p^2 = 0.17$, Figure 7). Regarding *Distractor* $\times M \times V$, when *V* was small or *M* was large, the difference between *Distractor* was significant. Regarding $A \times M \times V$, decreasing *M* or increasing *V* decreased *DT*.

### Movement Time
We observed the main effects for *A* ($F_{1,11} = 90.18, p < 0.001, \eta_p^2 = 0.89$) and *M* ($F_{3,33} = 326.65, p < 0.001, \eta_p^2 = 0.97$) not *Distractor* ($F_{1,11} = 1.14, p = 0.31, \eta_p^2 = 0.09$) and *V* ($F_{3,33} = 2.49, p = 0.08, \eta_p^2 = 0.18$). Figure 8 shows the results of the post-hoc test. We also observed the interactions for *Distractor* $\times M$ ($F_{3,33} = 6.01, p < 0.01, \eta_p^2 = 0.35$), $M \times V$ ($F_{9,99} = 4.80, p < 0.001, \eta_p^2 = 0.30$), and *Distractor* $\times M \times V$ ($F_{9,99} = 2.56, p < 0.05, \eta_p^2 = 0.19$). For *Distractor* $\times M \times V$, the difference between *Distractor* was not significant, increasing *M* decreased *MT*, and increasing *V* slightly decreased *MT* (Figure 9).

### Standard Deviation of x-coordinate
We observed the main effects for *Distractor* ($F_{1,11} = 31.49, p < 0.001, \eta_p^2 = 0.74$), *M* ($F_{3,33} = 95.20, p < 0.001, \eta_p^2 = 0.90$), and *V* ($F_{3,33} = 2.98, p < 0.05, \eta_p^2 = 0.21$) not *A* ($F_{1,11} = 0.72, p = 0.41, \eta_p^2 = 0.06$). Figure 10 shows the results of the post-hoc test. We also observed the interactions for *Distractor* $\times M$ ($F_{3,33} = 25.67, p < 0.001, \eta_p^2 = 0.70$), *Distractor* $\times V$ ($F_{3,33} = 4.75, p < 0.01, \eta_p^2 = 0.30$), and *Distractor* $\times M \times V$ ($F_{9,99} = 2.98, p < 0.01, \eta_p^2 = 0.21$). For *Distractor* $\times M \times V$, when *M* was large, the difference between *Distractor* was significant (Figure 11).

### Error Rate
We observed the main effects for *M* ($F_{3,33} = 23.39, p < 0.001, \eta_p^2 = 0.68$) and *V* ($F_{3,33} = 3.20, p < 0.05, \eta_p^2 = 0.23$) not *Distractor* ($F_{1,11} = 0.00, p = 0.95, \eta_p^2 = 0.00$) and *A* ($F_{1,11} = 0.24, p = 0.63, \eta_p^2 = 0.02$). Figure 12 shows the results of the post-hoc test. No interactions were observed.

### Model Fitting
Although there was no significant difference between *Distractor* conditions, we decided to verify the model fitness separated by each *Distractor*; models do not include the variable of *Distractor*. The reason is that interfaces simulated by the task differed depending on the presence or absence of the distractors. In addition, we believe that it may be inconvenient for models to include *Distractor* because even if the absence of the distractors is predicted to decrease movement time, UI designers cannot remove the distractors from a navigation bar for example.

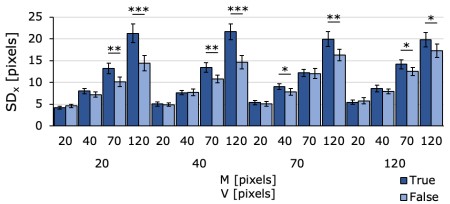

**Figure 11.** *SD$_x$* **vs.** *Distractor* $\times M \times V$.

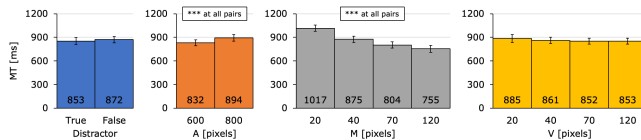

**Figure 8.** *MT* **vs.** *Distractor*, *A*, *M*, **and** *V*.

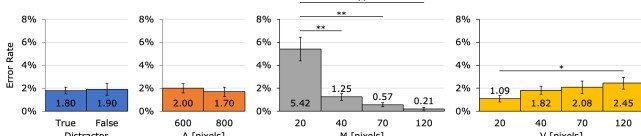

**Figure 12. Error rate vs.** *Distractor*, *A*, *M*, **and** *V*.

We found that movement time was affected significantly by motor width and slightly by visual width (Figure 8). Thus, following previous studies [35, 36], we selected $ID_m$ (Models #1 and #2 in Table 1) and $ID_v$ (Models #3 and #4 in Table 1) as candidate models. These models are built by replacing $W$ in the original Fitts' law (Equations 3 and 4) with the motor ($M$) or visual ($V$) width. As shown in Table 1, the $ID_m$ models showed sufficient fits for each *Distractor*. However, we also found that visual width affected the spread of clicked positions and error rate (Figures 10 and 12). Thus, interfaces designed on the basis of $ID_m$ (i.e., only considering that motor width) may frustrate users when they perform pointing operations. That is, we believe that a model needs to include the visual width.

When users perform operations on interfaces where motor and visual widths are different, they can only aim for the visual width at first, see the motor width by highlighting it, and then operate a cursor on the basis of the motor width. Based on the results that the interactions for $A \times M$ and $A \times V$ were not observed and that increasing visual width slightly decreases movement time, visual width may be added to a model in a form similar to motor width. Our model is as follows:

$$MT = a + b \log_2 \left( \frac{A}{M} \right) + c \log_2 \left( \frac{A}{V} \right). \qquad (10)$$

In normal Fitts' tasks, motor width equals visual width (i.e., $M = V$), and when $M = V$, letting $b' = b + c$, our model equals the original Fitts' law (Equation 3). In addition, Equation 10 can be converted as follows:

$$MT = a + b \log_2 \sqrt{ \left( \frac{A}{M} \right)^2 + c \left( \frac{A}{V} \right)^2 }. \qquad (11)$$

According to Hoffmann et al. [26], if $c$ is smaller than $b$, Equation 10 can be approximated to Equation 11. Note that Equation 11 was not derived to account for the weighted Euclidean distance of the target width and height proposed in a previous study [2]. We found that the effect of $V$ on movement time is smaller than that of $M$ ($\eta_p^2 = 0.18$ vs. 0.97, respectively); thus, we believe that $c$ may also become small, enabling approximation. Equation 11, when $M = V$, letting $a' = a + b \log_2 \sqrt{(1 + c)}$, equals Equation 3. That is, our two models have consistency with the original Fitts' law.

In addition to these models, we verified the Shannon formulation versions of the models (Models #6 and #8 in Table 1) where '+1' is added to the logarithm term of the models. It was revealed that '+1' improved model fitness [21, 25, 26, 28, 29]; thus, we added the '+1' versions to the candidate models. Such a posteriori modification has been conducted previously, e.g., [12].

Table 1 shows all candidate models. Some models have two regression constants, and others have three. Comparing $ID_{m1}$ and $ID_{mv1}$ (segmented) for example, when $c = 0$ in $ID_{mv1}$ (segmented), these models become the same, i.e., $ID_{mv1}$ (segmented) shows a better $R^2$ than $ID_{m1}$. Thus, we analyzed the model fitness by using adjusted $R^2$ and Akaike's Information Criterion (*AIC*) [5]. A model that shows a good fit shows a higher adjusted $R^2$ and lower *AIC* [15, 32, 43]. As shown in Table 1, for both *Distractor* conditions, $ID_{mv1}$ (combined) and $ID_{mv2}$ (combined) showed the best fits. In addition, the $c$ values in our models were small, which is consistent with the slight effect of $V$.

Because movement time strongly depended on motor width and increasing motor width increased $SD_x$, we verified the model fitness of $ID_e$ (Equation 5) by each *Distractor*. Under both *Distractor* conditions, $ID_e$ showed sufficient fits ($MT = 262 + 146 ID_e$ with $R^2 = 0.937$ in *Distractor* = *True*, $MT = 241 + 150 ID_e$ with $R^2 = 0.858$ in *Distractor* = *False*). Usuba et al. [35] also showed that using the effective width allows researchers to predict movement time, and our results support this fact. We also found that the effective width can be used even if the distractors exist. That is, researchers can compare the performance of devices (e.g., mouse vs. finger) with different accuracies under a condition in which motor and visual widths differ and distractors exist.

**Discussion**

As shown in Figure 8, we found that $MT$ did not depend on the presence or absence of the distractors (*Distractor*). Some participants said that they always aimed for the center of the target regardless of whether the distractors existed. This is one reason that the distractors did not affect movement time. On the other hand, the spread of clicked positions ($SD_x$) was affected by *Distractor* (Figure 10): the presence of the distractors increased $SD_x$. Some participants said that they performed pointing operations while relying on the highlight of the motor width of the left distractor; they judged the size of the motor width of the target by observing the highlight of the motor width of the left distractor. Thus, the participants sometimes accidentally clicked on the motor width of the distractor. We believe that such operation increased $SD_x$.

Usuba el al. found that dwell time and movement time are U-shaped functions whose origin point is located where the motor and visual widths are the same when users click on a target with different motor and visual widths [35]. As shown in Figures 7 and 9, we did not obtain such results. In their tasks, the motor width was highlighted before starting a trial, i.e., the participants knew the motor width in advance. In our task, the participants did not know the motor width in advance; thus, we believe this is why different results were obtained

For *Distractor* = *True*, $ID_{m2}$ showed a good fit; however, for *Distractor* = *False*, because the model fitness of $ID_{m2}$ decreased, we found that the effect of $V$ should be considered. The model fitness of $ID_{mv2}$ (combined) showed the best fits under both *Distractor*. In addition, comparing with $ID_{mv1}$ (combined), we also found that adding '+1' improves model fitness even if motor and visual widths are different.

Table 1. Model fitting by each *Distractor* ($N = 32$). All regression constants *a*, *b*, and *c* with 95% confidence intervals (CIs) [lower, upper].

| Model | Equation | Distractor = True a | b | c | adj. $R^2$ | AIC | Distractor = False a | b | c | adj. $R^2$ | AIC |
|---|---|---|---|---|---|---|---|---|---|---|---|
| #1 $ID_{m1}$ | $MT = a + b\log_2\left(\frac{A}{M}\right)$ | 431 [394, 470] | 112 [102, 122] | | 0.948 | 306 | 515 [461, 569] | 95.0 [81.1, 109] | | 0.872 | 328 |
| #2 $ID_{m2}$ | $MT = a + b\log_2\left(\frac{A}{M} + 1\right)$ | 382 [341, 423] | 121 [111, 131] | | 0.953 | 304 | 472 [414, 530] | 103 [88.3, 17] | | 0.879 | 326 |
| #3 $ID_{v1}$ | $MT = a + b\log_2\left(\frac{A}{V}\right)$ | 791 [622, 959] | 16.6 [-26.6, 59.8] | | 0.021 | 401 | 794 [647, 941] | 20.9 [-16.9, 58.7] | | 0.042 | 392 |
| #4 $ID_{v2}$ | $MT = a + b\log_2\left(\frac{A}{V} + 1\right)$ | 783 [597, 969] | 17.9 [-28.7, 64.5] | | 0.021 | 401 | 785 [622, 948] | 22.6 [-18.2, 63.3] | | 0.042 | 392 |
| #5 $ID_{mv1}$ (segmented) | $MT = a + b\log_2\left(\frac{A}{M}\right) + c\log_2\left(\frac{A}{V}\right)$ | 390 [343, 437] | 111 [102, 120] | 11.6 [2.69, 20.5] | 0.959 | 301 | 455 [389, 520] | 94.2 [81.9, 107] | 16.7 [4.34, 29.0] | 0.898 | 322 |
| #6 $ID_{mv2}$ (segmented) | $MT = a + b\log_2\left(\frac{A}{M} + 1\right) + c\log_2\left(\frac{A}{V} + 1\right)$ | 335 [286, 385] | 120 [111, 129] | 12.6 [3.52, 21.7] | 0.963 | 298 | 405 [335, 475] | 102 [89.3, 115] | 18.1 [5.27, 30.9] | 0.906 | 320 |
| #7 $ID_{mv1}$ (combined) | $MT = a + b\log_2\sqrt{\left(\frac{A}{M}\right)^2 + c\left(\frac{A}{V}\right)^2}$ | 387 [347, 426] | 121 [112, 130] | 0.035 [0.011, 0.058] | 0.967 | 295 | 427 [378, 476] | 112 [101, 124] | 0.102 [0.047, 0.157] | 0.942 | 304 |
| #8 $ID_{mv2}$ (combined) | $MT = a + b\log_2\left(\sqrt{\left(\frac{A}{M}\right)^2 + c\left(\frac{A}{V}\right)^2} + 1\right)$ | 336 [295, 377] | 130 [121, 140] | 0.036 [0.012, 0.059] | 0.970 | 291 | 381 [330, 431] | 121 [109, 132] | 0.104 [0.051, 0.158] | 0.947 | 301 |

In summary, we recommend UI designers to use $ID_{mv2}$ (combined). The time prediction model for different motor and visual widths was built for the first time from our experiment, and our results extended the knowledge of previous studies.

## EXPERIMENT 2: EFFECTS OF INTERVAL BETWEEN DISTRACTORS AND TARGET

Navigation bars sometimes have intervals between items such as in Figure 1b. When $M > V$ in Figure 5, there seem to be intervals; however, the motor width of the target touches those of the distractors. In Experiment 2, the target does not touch the distractors in either motor or visual width (Figure 2c). In Experiment 1, when there were intervals, the participants could predict that the motor width is larger than the visual width. However, in Experiment 2, because there were intervals between the motor widths, participants could not predict it. On the basis of Equation 9, we presumed that increasing the intervals decreased the movement time; that is, we presumed that pointing performance depends on the size of the intervals.

The apparatus, participants, and measurements were the same as in Experiment 1.

### Task, Design, and Procedure

In this experiment, the task (Figure 2c) included the intervals ($I$) between the target and distractors in addition to the task of Experiment 1. The participants had to do the same actions as in Experiment 1: they clicked on the blue start area and then aimed for the green target while avoiding the white distractors.

The variables of $A$, $M$, and $V$ were the same as those in Experiment 1. Unlike in Experiment 1, there were always the distractors (i.e., always *Distractor = True*). The $I$ was 0, 20, 40, or 70 pixels (0, 3.41, 6.82, or 11.94 mm, respectively)

The orders of $A$, $M$, $V$, and $I$ were randomized. One *set* consisted of $2A \times 4M \times 4V \times 4I = 128$ trials. After an introductory practice set, each participant completed seven sets to produce experimental data. A total of 10,752 trials (i.e., $2A \times 4M \times 4V \times 4I \times 7$ sets $\times 12$ participants) were conducted, which required approximately 35 min per participant.

### RESULTS

Among the 10,750 trials (there were 2 outliers), 382 errors occurred (3.55%). We analyzed the data by using repeated-measures ANOVA with Bonferroni correction as the $p$-value

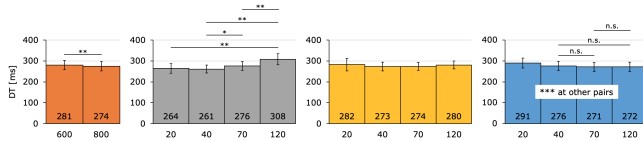

**Figure 13.** $DT$ vs. $A$, $M$, $V$, and $I$.

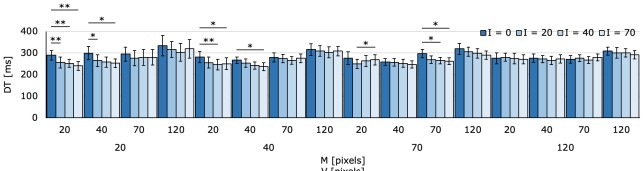

**Figure 14.** $M \times V \times I$ for $DT$.

adjustment method. The independent variables were $A$, $M$, $V$, and $I$, and the dependent variables were the same as those in Experiment 1.

### Dwell Time

We observed the main effects for $A$ ($F_{1,11} = 14.57, p < 0.01, \eta_p^2 = 0.57$), $M$ ($F_{3,33} = 21.58, p < 0.001, \eta_p^2 = 0.66$), and $I$ ($F_{3,33} = 33.01, p < 0.001, \eta_p^2 = 0.75$) not $V$ ($F_{3,33} = 0.62, p = 0.61, \eta_p^2 = 0.053$). Figure 13 shows the results of the post-hoc test. We also observed the interactions for $V \times I$ ($F_{9,99} = 4.27, p < 0.001, \eta_p^2 = 0.28$) and $M \times V \times I$ ($F_{27,297} = 1.78, p < 0.05, \eta_p^2 = 0.14$). When $M$ and $V$ were small for $M \times V \times I$, the differences between $I$s were significant (Figure 14).

### Movement Time

We observed the main effects for $A$ ($F_{1,11} = 114.77, p < 0.001, \eta_p^2 = 0.91$), $M$ ($F_{3,33} = 160.12, p < 0.001, \eta_p^2 = 0.94$), and $I$ ($F_{3,33} = 4.57, p < 0.01, \eta_p^2 = 0.29$) not $V$ ($F_{3,33} = 2.33, p = 0.092, \eta_p^2 = 0.17$). Figure 15 shows the results of the post-hoc test. We also observed the interactions for $M \times V$ ($F_{9,99} = 5.42, p < 0.001, \eta_p^2 = 0.33$) and $V \times I$ ($F_{9,99} = 2.34, p < 0.05, \eta_p^2 = 0.18$). For $M \times V$, increasing $M$ increased the differences between $V$s (Figure 16 left). For $V \times I$, when $V = 40$, the differences between $I$s were significant (Figure 16 right).

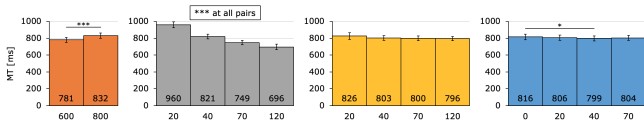

**Figure 15.** *MT* vs. *A*, *M*, *V*, and *I*.

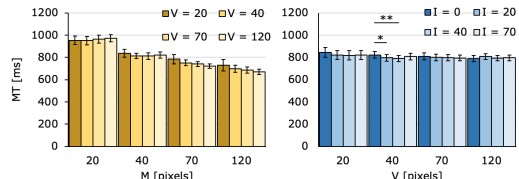

**Figure 16.** $M \times V$ and $V \times I$ for *MT*.

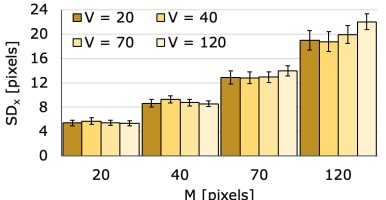

**Figure 18.** $M \times V$ for $SD_x$.

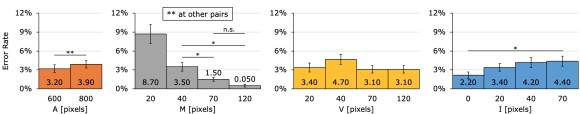

**Figure 19.** Error rate vs. *A*, *M*, *V*, and *I*.

### Standard Deviation of x-coordinate

We observed the main effects for *M* ($F_{3,33} = 136.63, p < 0.001, \eta_p^2 = 0.93$), *V* ($F_{3,33} = 3.12, p < 0.05, \eta_p^2 = 0.22$), and *I* ($F_{3,33} = 3.04, p < 0.05, \eta_p^2 = 0.22$) not *A* ($F_{1,11} = 2.81, p = 0.12, \eta_p^2 = 0.20$). Figure 17 shows the results of the post-hoc test. We also observed the interaction for $M \times V$ ($F_{9,99} = 3.49, p < 0.001, \eta_p^2 = 0.24$). Regarding $M \times V$, increasing *M* increased the differences between *V* (Figure 18).

### Error Rate

We observed the main effects for *A* ($F_{1,11} = 7.89, p < 0.05, \eta_p^2 = 0.42$), *M* ($F_{3,33} = 24.72, p < 0.001, \eta_p^2 = 0.69$), *V* ($F_{3,33} = 3.53, p < 0.05, \eta_p^2 = 0.24$) and *I* ($F_{3,33} = 5.66, p < 0.01, \eta_p^2 = 0.34$). Figure 19 shows the results of the post-hoc test. We also observed the interactions for $M \times I$ ($F_{9,99} = 2.04, p < 0.05, \eta_p^2 = 0.16$) and $A \times M \times I$ ($F_{9,99} = 2.50, p < 0.05, \eta_p^2 = 0.18$). For $A \times M \times I$, increasing the size of *I* almost always increased the error rate (Figure 20).

### Model Fitting

As shown in Figure 15, we found that increasing the *I* between the target and distractors decreased *MT*. We also found the interactions for $M \times V$ and $V \times I$ on *MT*. In addition, we observed that *M* and *I* had larger effects than *V*. Thus, we presume that the relationship between *M* and *V* is similar to that between *I* and *V*.

We can obtain a model (Equation 12) by adding *I* in a similar form to *M* to Model #8 in Table 1. The range of *I* is presumed to be $[0, \infty]$; thus, if we simply add *I* to the model, division by zero occurs when $I = 0$. On the basis of a previous study [33], we added '0.0049', which can be rounded to '0.00' due to preventing division by zero.

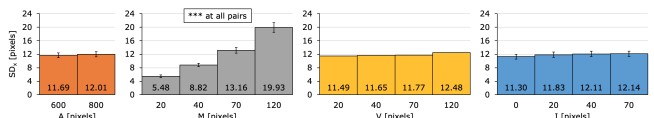

**Figure 17.** $SD_x$ vs. *A*, *M*, *V*, and *I*.

$$MT = a + b_1 \log_2\left(\sqrt{\left(\frac{A}{M}\right)^2 + c\left(\frac{A}{V}\right)^2} + 1\right) + b_2 \log_2\left(\sqrt{\left(\frac{A}{I+0.0049}\right)^2 + d\left(\frac{A}{V}\right)^2} + 1\right) \quad (12)$$

Hereafter, $b_1$, $b_2$, and *d* are also regression constants.

Considering that the effect of *V* was slight, the term $d\left(\frac{A}{V}\right)$ may not contribute much to the model fitness. In addition, because the interaction for $A \times I$ was not observed and it is convenient that the model is consistent with Model #8 in Table 1, Equation 12 is converted as follows.

$$MT = a + b_1 \log_2\left(\sqrt{\left(\frac{A}{M}\right)^2 + c\left(\frac{A}{V}\right)^2} + 1\right) + b_2 \log_2\left(\frac{1}{I+0.0049} + 1\right) \quad (13)$$

In this Equation, when $I = 0$ for example, the logarithm term including *I* becomes $b_2 \log_2(1/0.0049 + 1)$; thus, it is a constant. In addition, when $I = \infty$, the logarithm term becomes $b_2 \log_2(1)$; thus, it vanishes. That is, Equation 13 is consistent with Model #8 in Table 1. Moreover, in the original Fitts' task where $M = V$ and $I = \infty$ (there are no distractors), Equation 13 can be approximated to Equation 4. Although Equation 9 can consider the position of the distractors, all distractors need to have the same *ID* as that of the target. In Experiment 2, the distractors have the same motor and visual width as that of the target, i.e., the distractors' *ID* differ from the target's *ID*. Thus, we newly built the model that can consider the interval between the target and distractors instead of Equation 9.

We verified adjusted $R^2$ and *AIC* of all candidate models (Table 2). In the candidate models, we used '+1' versions. The $ID_{mvi2}$ model showed the highest $R^2$ and lowest *AIC*. The difference between the *AIC* values of $ID_{mvi1}$ and $ID_{mvi2}$ was small. However, because $ID_{mvi2}$ has fewer constants and is

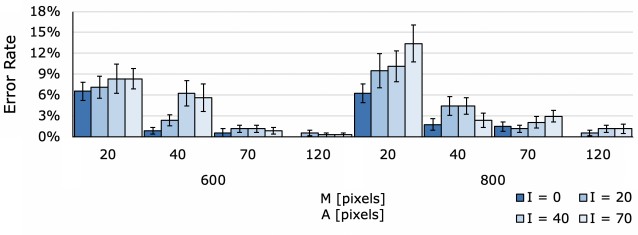

**Figure 20.** Error rate vs. $A \times M \times I$.

**Table 2. Model fitting for all conditions ($N = 128$). All regression constants $a$, $b_1$, $c$, $d$, and $b_2$ with 95% CIs [lower, upper].**

| | Model | Equation | $a$ | $b_1$ | $c$ | $d$ | $b_2$ | adj. $R^2$ | AIC |
|---|---|---|---|---|---|---|---|---|---|
| #1 | $ID_{m2}$ | $MT = a + b_1 \log_2\left(\frac{A}{M} + 1\right)$ | 374 [351, 397] | 111 [105, 117] | | | | 0.924 | 1234 |
| #2 | $ID_{mv2}$ (combined) | $MT = a + b_1 \log_2\left(\sqrt{\left(\frac{A}{M}\right)^2 + c\left(\frac{A}{V}\right)^2} + 1\right)$ | 314 [292, 337] | 123 [118, 128] | 0.054 [0.038, 0.071] | | | 0.954 | 1172 |
| #3 | $ID_{mvi1}$ | $MT = a + b_1 \log_2\left(\sqrt{\left(\frac{A}{M}\right)^2 + c\left(\frac{A}{V}\right)^2} + 1\right) + b_2 \log_2\left(\sqrt{\left(\frac{A}{I+0.0049}\right)^2 + d\left(\frac{A}{V}\right)^2} + 1\right)$ | 307 [284, 329] | 123 [118, 128] | 0.055 [0.038, 0.072] | -0.082 [-0.082, -0.082] | 0.99 [0.28, 1.69] | 0.957 | 1166 |
| #4 | $ID_{mvi2}$ | $MT = a + b_1 \log_2\left(\sqrt{\left(\frac{A}{M}\right)^2 + c\left(\frac{A}{V}\right)^2} + 1\right) + b_2 \log_2\left(\frac{1}{I+0.0049} + 1\right)$ | 311 [289, 333] | 123 [118, 128] | 0.054 [0.038, 0.070] | | 1.74 [0.56, 2.92] | 0.957 | 1165 |

**Table 3. Model fitting for three experiments of previous studies. All regression constants $a$, $b$, and $c$ with 95% CIs [lower, upper].**

| | Model | Equation | Experiment in [36] ($N = 24$) | | | | | Experiment 1 in [35] ($N = 32$) | | | | | Experiment 2 in [35] ($N = 32$) | | | | |
|---|---|---|---|---|---|---|---|---|---|---|---|---|---|---|---|---|---|
| | | | $a$ | $b$ | $c$ | adj. $R^2$ | AIC | $a$ | $b$ | $c$ | adj. $R^2$ | AIC | $a$ | $b$ | $c$ | adj. $R^2$ | AIC |
| #1 | $ID_{m2}$ | $MT = a + b \log_2\left(\frac{A}{M} + 1\right)$ | 263 [111, 414] | 159 [134, 183] | | 0.901 | 258 | 486 [417, 554] | 112 [94.6, 129] | | 0.861 | 336 | 381 [356, 406] | 115 [109, 121] | | 0.975 | 291 |
| #2 | $ID_{mv2}$ (combined) | $MT = a + b \log_2\left(\sqrt{\left(\frac{A}{M}\right)^2 + c\left(\frac{A}{V}\right)^2} + 1\right)$ | 140 [-74.6, 103] | 192 [179, 205] | 0.0086 [0.0061, 0.011] | 0.980 | 222 | 399 [329, 469] | 129 [113, 145] | 0.087 [0.025, 0.15] | 0.914 | 323 | 362 [331, 393] | 117 [111, 123] | 0.16 [-0.033, 0.36] | 0.977 | 291 |

consistent with Model #8 in Table 1 and Equation 4, $ID_{mvi2}$ is the best of the candidate models.

The model fitness of $ID_e$ ($N = 128$) was $MT = 229 + 145ID_e$ with $R^2 = 0.870$; thus, using the effective width also showed a sufficient fit in Experiment 2.

**Discussion**

As shown in Figure 15, we found that enlarging $I$ between the target and distractors decreased $MT$. However, enlarging $I$ increased the spread of the clicked positions ($SD_x$) and error rate (Figures 17 and 19). Thus, a UI designer can provide wide intervals to allow users to perform operations more quickly but less accurately. In addition, the dwell time was not a U-shaped function. This is not consistent with that found by Usuba et al. [35] but matches that from Experiment 1. That is, when users do not know the size of the motor width in advance, the dwell time may not be a U-shaped function.

We constructed a model (Model #4 in Table 2) that can consider the intervals between the target and distractors, and the model showed the highest adjusted $R^2$ and lowest $AIC$. In terms of the equation form, the model is consistent with Model #8 in Table 1 constructed on the basis of the results from Experiment 1. However, the predicted $MT$ may not be consistent with the results of Experiment 1. In Experiment 1, as shown in Figure 8, although the difference between $Distractor$ was not significant for $MT$, $MT$ when $Distractor = True$ was smaller than that when $Distractor = False$. However, according to Equation 13, the predicted $MT$ when $Distractor = True$ (i.e., $I = 0$) was larger than that when $Distractor = False$ (i.e., $I = \infty$). Thus, we believe that to apply our model to wide ranges of conditions, it should be refined. On the other hand, comparing Model #4 in Table 2 with Equation 9, both models show that increasing the interval between the target and distractors decreases the movement time. That is, enlarging $I$ does not necessarily decrease $MT$; thus, we believe that there is a threshold for the effect of $I$.

In summary, although there are some limitations, we constructed a model that can consider the difference between motor and visual widths and the intervals between the target and distractors on the basis of the results from Experiments 1 and 2.

**MODEL FITTING FOR DATA OF EXISTING STUDIES**

Usuba et al. also conducted experiments in which participants clicked on the target with different motor and visual widths; one experiment involved a small target width [36] and the others involved a larger one [35]. We verified whether our model, i.e., $ID_{mv2}$ (combined), shows a good fit for their data. Their studies also found that movement time is affected strongly by motor width and slightly by visual width. The results of their studies are similar to ours; thus, we believe that $ID_{mv2}$ (combined) can predict movement time for their data more accurately.

Table 3 shows the model fitness for Experiment in [36], Experiment 1 in [35], and Experiment 2 in [35]. Except for the data of Experiment 2 in [35], $ID_{mv2}$ (combined) showed larger $R^2$ and lower $AIC$. In Experiment 2 in [35], the range of motor width by each visual width depended on the value of the visual width. As Usuba et al. mentioned in that paper, because the effect of visual width depended on the range of the motor width (the effect of visual width decreased), the original Fitts' law showed high $R^2$. Thus, the results may also depend on the experimental condition. On the other hand, considering the difference in $AIC$, $ID_{mv2}$ (combined) did not show a worse fit for Experiment 2 in [35]. In summary, $ID_{mv2}$ (combined) showed good fits for the data of three previous studies, which empirically support it.

**LIMITATION AND FUTURE WORK**

Our $ID_{mvi2}$ (combined) can provide UI designers with the optimal motor width, visual width, and intervals in terms of movement time; however, we did not find the optimal values in terms of total user performance. In Experiments 1 and 2, increasing the visual width or increasing the intervals increased the error rate; however, our model shows the opposite; increasing the visual width or increasing intervals decreased movement time. That is, in interfaces based on our model, users can perform pointing operations faster but may become frustrated. In addition, if a navigation bar has larger intervals between items, the navigation bar also becomes larger; the distance to each item is also larger, and the total movement time in the navigation bar may become longer. Constructing a model considering total user performance is for future work.

Our model showed good fits for five experimental datasets (two internal dataset and three external dataset), which supports

that its high fitness is not overfitting. That is, our model was shown to be empirically correct.

In actual GUIs, such as those in Figure 1, for example, the positions of distractors are vertical or horizontal, the visual width by each object differs, and the objects have certain height. Our model is a baseline model and for 1D pointing tasks. More practically, it should be refined to consider the above factors.

## CONCLUSION

We conducted two experiments to investigate the effect of distractors and intervals between the target and distractors. On the basis of the results, we constructed a model that can consider motor and visual widths and intervals. Our model shows good fits for not only the data of our two experiments but also those of three previous studies. That is, it allows designers to adjust motor and visual widths and intervals on the basis of the movement time. We also found that even when there are distractors and intervals, Fitts' law using the effective width shows a good fit. Thus, researchers can also compare input devices that have different accuracy in such a situation. We expect that in modeling studies, including ours, any pointing operations and situations can be modeled, and all users can explore GUIs without frustration.

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
