# OpenReview forum: "Time Prediction Model for Pointing at Target Having Different Motor and Visual Width with Distractors"
_graphicsinterface.org/Graphics_Interface/2020/Conference — Submitted to GI 2020_

### Official Review · AnonReviewer3 · 2020-04-20
**This paper runs two variations of the standard Fitts' law study...**

**Rating:** 5
**Confidence:** 4

**Review:**

This paper presents two variations of the standard Fitts' law study, to understand the effect of (1) a situation where targets initially appear with a given size (called the "visual width" in the paper) but are revealed to have a larger clickable size revealed once the cursor gets close (called the "motor width") or vice versa; and (2) different gaps between targets arranged side-by-side. Models are fit which account for these differences, on both new data gathered from 12 participants, and data sets gathered from several past studies.

Overall, I found the design of the study to be sound, as is the data analysis and modeling methodology. I also think that the overall motivation of understanding whether interfaces with distinct visual and motor widths (to use the paper's terms) is interesting.

Despite the above, I am not very enthusiastic about this paper. While I appreciate the overall motivation, I'm not sure if a Fitts' law study is the right approach for going about understanding the effects of these kinds of interfaces. Or, put in a different way, I'm not sure if the study results are all that valuable for designers (given that it's looking at 1D pointing), or whether this type of interface is common enough that it's useful to have a new Fitts' law formula to account for it. The situation in which motor width differs from visual width seems fairly niche overall, and the examples cited in the introduction where visual width is greather than motor width seems like a situation that will almost always be due to poor interface implementation, rather than a conscious design decision.

In addition to the above concerns about the contribution of the paper, the term "motor size" is already used in Blanch et al.'s CHI 2004 work to refer to the situation where the control-display gain is manipulated to create objects with a higher or lower size in motor space as compared to their visual space on screen, work which is not cited in this paper. It seems awkward to use such a similar term here, when C-D manipulation is not the focus.

Finally, I found the study results to be difficult to interpret, as many of the results subsections are ANOVA output with little interpretation and commentary to help the reader understand what was found.

Based on the above, I feel the paper is marginally below the acceptance threshold.

---

### Official Review · AnonReviewer2 · 2020-04-21
**Writing issues. Not ready for publication**

**Rating:** 4
**Confidence:** 4

**Review:**

This paper reports models for predicting pointing time on targets that have a different visual and motor width. Visual width is how the targets visually appears to users, and motor width is the actual areas on which users can acquire the target. The goal of the paper is to provide an all in one Fitts like model, factoring in amplitude, visual width, motor width, as well as interval between the target and potential adjacent distractors. Two experiments varying these factors were conducted.

I would like to start my review by thanking the authors for their paper. It is an interesting piece of work that offered new perspectives on approaching pointing tasks.
That being said, I'm afraid the current state of the writing prevent me from arguing for acceptance.

Throughout my reading, I struggled with the concept definitions. Although defined, at the beginning, a clear context is lacking (on which situations the difference between motor and visual width appears in). In particular for situation where there is a the shorter motor width, which seems uncommon. Figure 1 provides an example of it, but it seems that it is a UI/ergonomic problem which deems the need for a model describing this situation. I understand my comment might not be relevant, as one could argue that we could still study a broader picture. However, in several papers from the related work [35,36], compelling situation (like resizing windows) are given. Therefore I do believe, reshaping the introduction using these examples early on would sweep away doubts and get the reader on board from the get-go. I would also shorten drastically the first paragraph to get quicker to the "meat".

Neat related work.

For the study design, only one repetition of each factor combination for each participant seems really odd. I understand the need to keep the xp time under 45min, but could 1M and 1V could have been dropped? Also why not choosing amplitudes that were drastically different? And having only one distractor on either side with drastically different color schemes seems quite far from realistic settings. One could argue this helped participants to simply ignore distractors.

Apart from, the ANOVA results, no figures are reported in the results which makes it hard to grasp which hypothesis could be drawn or not. Given the size of the graphs and the difficulty to read them, a lot of effort is required to jump back and forth from graphs to results. I struggled interpreting the results, and was left having to decide whether I trust or not the authors assessments, which is generally not a good sign. Similarly, curves (eg. "U" shapes) are being referred to and compared to related work ones, however, never presented to the reader.

The model fitting parts are very straightforward and tend to be text explanation of equation rather than explanation of rational. A lot of information are given on 8 variations of a model, with a clear presentation in the end that only one or two can be kept. Since all models are coming from the authors themselves, there is no particular need for comparison. I understand the method used by the authors could have been to test every possible candidates, but  I would argue, presenting the reasoning behind the last model as well as the results. But no need for intermediate models.

Another minor point. I understand the need for a '0.0049' but explanations are seriously lacking to justify the value as well. As it currently reads, I feels somehow parachuted in.

A neat step to compare to the related work xps was taken, how did the authors check their models with these studies? Did they contacted the authors to get the data?

I'm also not entirely sure how the model would benefit UI designers. A use case on how to use these results could be useful. "Our IDmvi2 (combined) can provide UI designers with the optimal motor width, visual width, and intervals in terms of movement time; " What does it mean? Are UI designers fixing a time and then look at what sizes targets should be? If so, it seems pretty hard to use as is.

This papers presents models for predicting pointing time on targets that have a different visual and motor width. There is a value for the community. However the current state of the writing prevents me from arguing acceptance.

Minor comments:
 - English can be off sometimes, I would suggest getting the paper thoroughly re-read
 - "We observed the main effects for XX" -> "We observed main effects of XX"
 - "On the basis of a previous study" -> "Based on a previous study"
 - Titles of the graph figures are not explicit at all. I would suggest using full sentences to have self-explanatory captions.

---

### Official Review · AnonReviewer1 · 2020-04-21
**Very good fittings, but a lot to discuss and clarify.**

**Rating:** 5
**Confidence:** 4

**Review:**

This paper presents, develops and tests a series of models of pointing time in situations where the apparent size of a target does not represent its actual clickable area. The models include both visible and clickable widths in the visual space, as well as the distance between the clickable areas.
In two studies, the authors show that their models do better than classic Fitts' law. They also validate some of their models using data from similar past studies.

This paper is generally well written, the studies well described. The related work is clear and well documented. I particularly appreciated that the models are tested on prior data, even though the experimental setups seem very similar and therefore generalization remains an open question—still, few papers bother to do that.

However I also have a number of issues with this paper. In particular, I think the choice of experimental tasks needs to be justified before conclusions are drawn, as well as how these results would apply in real use. It is also unclear how designers could use these models in their work, which is left unexplained. Finally, a number of rationales need more justification to be convincing, and clarifications are required throughout the paper.


# APPLICABILITY

I feel like some discussions are missing to justify some of the design choices made in this work, and how it can be applied in real life as claimed.

## To Interface Design

Throughout the paper, the authors claim that these models will help interface designers. I propose that this is not an obvious outcome, and this paper would have benefited from e.g. a quick how-to.
Interaction designers already work under a number of constraints: label text lengths, available space, graphic charters, item positioning, readability, discoverability, aesthetics, etc. Some of these elements may share some parameters with the proposed models, but cannot simply become secondary to these models. With all that in mind, how can one integrate these models in a realistic design process?

## To Real Interfaces

This is mentioned a few times but never really addressed: how generalizable are the tasks performed in the presented studies? The examples provided at the beginning of the paper include mostly menus, for which one would expect that the difference between V and M would remain constant. How long can one use an interface where e.g. V = a+M, and not quickly get used to it? Yet the studies in this paper made this difference constantly changing and unknown, putting participants in a constant state of "discovering the interface". What scenario does that represent?
In that perspective, most of the presented results would only apply to the first few selections in a given targets set before its design flaws are understood and integrated by the user.

## To Realistic Mouse Interaction

>> "Following previous studies [14,15], we asked the participants to avoid clutching (replacing the mouse on the mouse pad)."
>> "We did this because clutching may reduce the fitness of pointing models [14]. If we had allowed clutching and obtained poor regression fitness, it would have been unclear whether the results were due to clutching or experimental conditions such as the difference between the motor and visual widths." (p. 4)
-
This seems like an argument for the sake of fitness reporting, not of empirical evidence. People do clutch quite a lot with mice in real life, so the presented results only apply to a subset of real-life pointing actions.Shouldn't a realistic model hold when clutching occurs? This is an artificial instruction that hurts the results more than they reveal an empirical truth.

## Should It Apply?

Overall I think there is something missing regarding the general phenomenon of having clickable objects that do not disclose their full shape, except perhaps in video games. There is probably a discussion to be had about whether this constitutes bad design (especially considering the "U-shape" of performance briefly mentioned in the paper), and in this case, whether this paper should provide guidelines to implement it.


# RATIONALES / REASONING

A number of rationales in this paper seem based on fragile bases, which makes me wonder if some of them are not just here to justify the models a posteriori.

For instance, it is entirely unclear to me why the transition between Equations (10) and (11) was at all necessary, at least _prior_ to performing the tests. Equation (11) is not mathematically equivalent to (10) and requires approximations, while (10) does not and was rather well justified. What made (11) an interesting model to explore beforehand?

>> "Usuba el al. found that dwell time and movement time are U-shaped functions whose origin point is located where the motor and visual widths are the same when users click on a target with different motor and visual widths [35]." (p. 6)
-
Several things here.
1) "origin" -> "minimum"
2) reformulate: either visual and motor widths are the same, or they are different
3) More importantly, knowing this, it appears strange to have suggested models in which the components involving V are simply added to the components involving M. A model of the form $log(a+x)$ or $log(sqrt(a^2+x^2))$, with $a, x > 0$, will increase with $x$ when $a$ is held constant, not form a U-shape around $x=a$. Of course, the quoted sentence does not specify the input parameter of that U-shape, so my interpretation could be wrong. This needs to be clarified, but otherwise it would seem that some informations were ignored when forming the model in Equation (10).

>> "As shown in Figure 15, we found that increasing the I between the target and distractors decreased MT." (p. 8)
>> "As shown in Figure 15, we found that enlarging I between the target and distractors decreased MT." (p. 9)
-
Fig. 15 shows a significant difference in *one* pair of I values, with a difference of perhaps 20 ms. It seems difficult to justify anything strongly, with such a specific and small difference. This 'result' is used twice in the paper, and it makes me question the arguments it introduces.

>> "However, enlarging I increased the spread of the clicked positions (SDx) and error rate (Figures 17 and 19)." (p. 9)
-
Fig. 17 shows no significant difference and a rather "flat" curve. Fig. 19 shows a significant difference in one pair, by about 2%. The arguments that follow therefore have brittle bases.

>> "We constructed a model (Model #4 in Table 2) that [...] showed the highest adjusted R2 and lowest AIC." (p. 9)
-
Compared to the 2nd and 3rd models in Table 2, the difference is 0 or 0.003 in adjR2, and 1 or 7 AIC units, which seems rather small to confidently select a model over another.

>> "Thus, a UI designer can provide wide intervals to allow users to perform operations more quickly" (p. 9)
-
Wide intervals can also mean more distance to the intended target, in certain configurations, which probably does not make pointing faster.


# CLARITY / INACCURACIES

Generally, "motor width" would be understood as "the size of the target in motor space", i.e. the distance a physical limb or device would have to travel to go through it (typically with fixed CD gain). In many interactive situations the target width in motor space is not equal to the area in which it can be clicked in the virtual space, whether that full visual width is visible or not. I believe other terms should be used throughout, e.g. "visible width" vs. "clickable width".

Conversely...
>> "Thus, changing the C-D gain allows users to feel as if the motor width is enlarged without changing the visual width." (p. 3)
...this sentence uses the "limb motor" definition of motor width, not the one used in the rest of the paper. With a different gain, this paper's definitions of visual and motor widths both change.

Throughout the paper, a number of r-squared values are deemed "sufficient". What is "sufficient", and why, should be clarified. R-squared are not one-size-fits-all metrics of goodness, their interpretation can be contextual.

Sentences like "a model for considering the difference between the motor and visual widths and the intervals between the target and distractors" (several times throughout the paper, in slightly different phrasings) also need to mention *what* is being modeled, i.e. pointing time.

>> "In graphical user interfaces (GUIs), users move a cursor and then click" (p. 1)
-
While prominent, this does not apply to all GUIs.

>> "passing through two goals called crossing [1]" (p. 1)
-
You mean "steering"?

Figure captions should be proofread.

>> "In normal Fitts’ tasks, the target has a certain width and practically infinite height, i.e., a 1D pointing task." (p. 3)
-
Define "normal." A large number of Fitts' pointing studies in HCI are performed on 2D and even 3D environments and target placement.

Figure 2 is referred to quite often, including near the end. This requires the reader to go back quite a lot, and quite often, while Figure 2 is not that necessary early in the paper.

Figure 4 should also show the feedback when M < V.

>> "The error rate was lower than those in previous studies [...]. participants[...] performed the pointing operation while watching the highlight of the motor width. Thus, we believe that because the highlight allows the participants to operate more accurately, a lower error rate was observed" (p. 4)
-
I do not understand, how else could participants have known the actual ("motor") size of the target anyway? Of course highlights make people more accurate, if there is no other way to tell where the real target is.
Also, for consistency and in relation to the point above, it would be interesting to also comment on possible differences in MT between studies, since many parameters were the same.

>> "On the basis of a previous study [33], we added ‘0.0049’, which can be rounded to ‘0.00’ due to preventing division by zero." (p. 8)
-
That value seems random in the absence of more detailed explanations, even if they were directly taken from [33].

---

### Meta-Review · Area_Chair1 · 2020-04-23

**Recommendation:** Reject
**Confidence:** 5

**Metareview:**

All reviewers found the paper interesting, with sound study design and methodology. However, the ratings are all somewhat negative, with various reasons:
- the paper is not very convincing as to why this phenomenon needs to be modeled at all (All Rs). The presented examples either feel "niche" or simply bad design, which the reviewers are not sure needs a model.
- it is unclear how exactly this model will benefit designers (All Rs), how it would integrate into their existing processes.
- R2 and R3 question the studies' approach and parameters.
- R2 and R3 offer comments on how to make the results more readable, and easier to interpret
- R1 and R3 criticize the use of the term "motor size/width", R1 and R2 feel like the 0.0049 value should be rationalized.

Other comments can be found in the individual reviews, all worthy of consideration for a future resubmission. In its current state, the consensus is to reject this submission until these issues are addressed.

---

### Decision · Program_Chairs · 2020-04-25

Reject